# Cardiovascular Safety of Hydroxychloroquine–Azithromycin in 424 COVID-19 Patients

**DOI:** 10.3390/medicina59050863

**Published:** 2023-04-29

**Authors:** Matthieu Million, Jean-Christophe Lagier, Jérôme Hourdain, Frédéric Franceschi, Jean-Claude Deharo, Philippe Parola, Philippe Brouqui

**Affiliations:** 1IHU-Méditerranée Infection, 13005 Marseille, France; jean-christophe.lagier@univ-amu.fr (J.-C.L.); philippe.parola@univ-amu.fr (P.P.); philippe.brouqui@univ-amu.fr (P.B.); 2Unité MEPHI (Microbes, Evolution, Phylogénies et Infection), Assistance Publique-Hôpitaux de Marseille, Institut de Recherche pour le Développement, Faculté des Sciences Médicales et Paramédicales, Aix Marseille University, 13005 Marseille, France; 3Assistance Publique—Hôpitaux de Marseille, Centre Hospitalier Universitaire La Timone, Service de Cardiologie, 13005 Marseille, France; jerome.hourdain@ap-hm.fr (J.H.); frederic.franceschi@ap-hm.fr (F.F.); jeanclaude.deharo@ap-hm.fr (J.-C.D.); 4C2VN, Faculté des Sciences Médicales et Paramédicales, Aix Marseille University, 13005 Marseille, France; 5VITROME, AP-HM, SSA, IRD, Faculté des Sciences Médicales et Paramédicales, Aix Marseille University, 13005 Marseille, France

**Keywords:** COVID-19, SARS-CoV-2, hydroxychloroquine, azithromycin, QTc interval, cardiac rhythm, safety, torsades de pointe

## Abstract

*Background and Objectives*: Hydroxychloroquine (HCQ) combined with azithromycin (AZM) has been widely administered to patients with COVID-19 despite scientific controversies. In particular, the potential of prolong cardiac repolarization when using this combination has been discussed. *Materials and Methods*: We report a pragmatic and simple safety approach which we implemented among the first patients treated for COVID-19 in our center in early 2020. Treatment contraindications were the presence of severe structural or electrical heart disease, baseline corrected QT interval (QTc) > 500 ms, hypokalemia, or other drugs prolonging QTc that could not be interrupted. Electrocardiogram and QTc was evaluated at admission and re-evaluated after 48 h of the initial prescription. *Results*: Among the 424 consecutive adult patients (mean age 46.3 ± 16.1 years; 216 women), 21.5% patients were followed in conventional wards and 78.5% in a day-care unit. A total of 11 patients (2.6%) had contraindications to the HCQ-AZ combination. In the remaining 413 treated patients, there were no arrhythmic events in any patient during the 10-day treatment regimen. QTc was slightly but statistically significantly prolonged by 3.75 ± 25.4 ms after 2 days of treatment (*p* = 0.003). QTc prolongation was particularly observed in female outpatients <65 years old without cardiovascular disease. Ten patients (2.4%) developed QTc prolongation > 60 ms, and none had QTc > 500 ms. *Conclusions*: This report does not aim to contribute to knowledge of the efficacy of treating COVID-19 with HCQ-AZ. However, it shows that a simple initial assessment of patient medical history, electrocardiogram (ECG), and kalemia identifies contraindicated patients and enables the safe treatment of COVID-19 patients with HCQ-AZ. QT-prolonging anti-infective drugs can be used safely in acute life-threatening infections, provided that a strict protocol and close collaboration between infectious disease specialists and rhythmologists are applied.

## 1. Introduction

By 10 March 2023, the SARS-CoV-2 outbreak had infected around 667 million people, and more than 6.7 million of COVID-19-related deaths had been reported [1]. Several specific candidate treatments have been tested in large randomized studies, but none has been globally recognized as the optimal treatment [2]. Early (<5 days of symptoms) oral treatment to prevent complications before they occur and death (nirmatrelvir/ritonavir) has been validated by the WHO only in 2022, i.e., two years after the pandemic emerged [3]. 

Starting in March 2020, our hospital department decided upon a strategy including treatment with hydroxychloroquine (HCQ) and azithromycin (AZ) for COVID-19 patients. This choice was supported by early Chinese publications about the antiviral effects of chloroquine (CQ) and its derivatives against SARS-CoV2; the demonstration of a synergistic effect in vitro of the HCQ-AZ combination on SARS-CoV-2; the HCQ and AZ immunomodulators effects, which may prevent the “cytokine storm” of COVID-19; the HCQ antithrombotic effects, which may also be useful in the context of COVID-19, associated with pulmonary embolism and coagulopathy; and the fact that HCQ-AZ has been associated with a reduction in viral shedding, with potential public health effects by reducing the duration of contagiousness [2]. AZ has the added advantage of preventing superinfection [2]. HCQ-AZM have been widely administered to patients, and observational studies with thousands of cases have been published throughout the world [4]. In our center in 2020, this combination was associated with lower mortality among 2111 COVID-19 hospitalized patients [5] and 10,429 COVID-19 outpatients [6]. 

The possible toxicity of HCQ or HCQ-AZ has been highlighted in published or retracted studies [7,8,9,10]. More specifically, HCQ-AZ combination has raised the question of a possible lengthening of the QT interval on electrocardiogram (ECG), which could lead to an increased risk of torsades de pointes and sudden death [11,12]. While initial cardiac safety publications evaluating HCQ treatment in hospitalized patients with COVID-19 showed significant QTc lengthening in some patients [13,14], another large study, evaluating HCQ safety in lower-risk patients, showed only a modest QTc prolongation without clinical consequences [15]. At this stage, monitoring of QTc had been suggested [16,17]. 

As the infectious disease team at our academic hospital had been involved in observing the role of HCQ-AZ in the treatment of COVID-19 infection early on, strict cardiac monitoring was immediately established during the initial national context of lockdown and limited medical resources. However, specific cardiovascular safety was not reported in our center. This is important to decipher if a strict standardized protocol for an association of QT-prolonging drugs allows their safe prescription for acute infectious diseases. Indeed, many important antimicrobial drugs are associated with prolonged QTc (Table 1). Here, we report the details and results of our Cardiac Rhythm Safety Strategy, which was composed of an initial clinical evaluation, followed by QTc monitoring in a cohort of 424 COVID-19 patients treated with HCQ-AZ.

## 2. Methods

The patients were the first adults seen for SARS-CoV-2 infection at the Institut Hospitalo-Universitaire (IHU) Méditerranée Infection, France, between 3 March 2020 and 5 April 2020. Our institute includes 75 hospital beds [18]. COVID-19 patients could be hospitalized in five different ways at our institute: (a) directly after screening at our day clinic; (b) outpatients initially followed at our day clinic and then requiring hospitalization; (c) from the emergency department; (d) from other hospital wards or nursing homes; and (e) from intensive care units. During this early period of COVID-19 care, even mild patients could have been hospitalized according to isolation request [19]. Clinical severity was assessed using the National Early Warning Score adapted to COVID-19 patients (NEWS-2) upon hospital admission, with three categories of clinical deterioration: low score (NEWS-2 = 0–4), medium score (NEWS-2 = 5–6), and high score (NEWS-2 ≥ 7) [20].

A systematic cardiac rhythm safety evaluation was performed before initiation of treatment using a simple assessment, as presented in Table 2 Briefly, medical history and current medical status were thoroughly assessed for each patient. Once the decision to treat with HCQ-AZ combination had been made by the ID clinicians, a 12-lead ECG was performed on each patient before treatment (baseline) and scheduled two days later after treatment had begun (Day 2). All ECGs were reviewed by senior cardiologists. Heart rate, QRS duration, and QT interval were measured, and the values automatically calculated by the recorder were collected. QTc was systematically calculated using the Bazett formula. No specific correction was made in cases of bundle branch block. Tracings were recorded at rest, with a paper speed of 25 mm/s and an amplitude calibration of 1 mm/mV (MAC^®^ 3500 or MAC^®^ 1600 recorder; GE Healthcare Europe, Freiburg, Germany). Treatment with HCQ and AZM was not started or was discontinued when the corrected QT interval (QTc; Bazett formula) was >500 ms and the risk–benefit ratio of HCQ and AZM was between 460 and 500 ms as estimated by an infectiologist and agreed upon by a cardiologist. Treatment was not started when the ECG showed patterns suggesting a channelopathy, and the risk–benefit ratio was discussed when it showed other significant abnormalities (i.e., pathological Q waves, left ventricular hypertrophy, and left bundle branch block). In addition, any drug with the potential of prolonging the QT interval was discontinued or replaced for the duration of treatment. Standard blood chemistry was checked, especially potassium levels. Any hypokalemia or hyperkalemia was corrected before the initiation of treatment. The infectious disease specialist was invited to contact a cardiologist whenever they felt the need, using a telephone “hot-line”, which was set-up in the emergency context of the pandemic. The drug regimen was as follows: HCQ at 200 mg 3 times a day for 10 days, plus AZ at 500 mg once a day on the first day, and then 250 mg once a day for 4 days during meal. All patients were physically seen or contacted by telephone on Day 11, i.e., one day after the end of the HCQ-AZM therapy.

The outpatients were informed of the need to contact the center in the event of unusual symptoms, including palpitations and syncope/dizziness, and were scheduled for an in-person follow-up appointment on Day 2 of treatment, and for remote follow-up afterward via the COVID AP-HM^®^ application (Version 1.0, RADHIUS SAS, Saint-Jean-d’Illac, France) or by telephone.

The quantitative variables are presented as means ± standard deviations, and the categorical variables are presented as numbers (percentages). An analysis was conducted in patients for whom both the baseline ECG and the Day 2 ECG were available. The initial QTc and Day 2 QTc were compared based on a paired *t*-test in the overall study sample and according to subgroups (age, sex, cardiopathy, and hospitalization). The predictive effect of various characteristics (age, heart rate, sex, and cardiopathy) on QTc prolongation ≥ 30 ms and ≥60 ms was assessed by estimating the odds ratios (ORs) with 95% confidence intervals (CIs).

An analysis was conducted in a random subsample of the overall study sample to compare the cardiologist’s interpretation and the automatic interpretation of the QTc. Correlation and agreement between the measures were assessed by estimating the correlation and the intraclass correlation coefficient (ICC), respectively, with 95% CI. All analyses were performed using the R software (version 3.6.3, The R Foundation for Statistical Computing, Vienna, Austria). All tests were two-sided, and *p*-values < 0.05 were considered to be statistically significant.

This study is a retrospective analysis of medical data collected during systematic cardiac rhythm safety evaluation performed before the initiation of a treatment that potentially prolonged the QTc interval to minimize cardiac consequences. The data were extracted from the patient medical files and then analyzed and stored according to the European General Protection of Data Regulation as we declared in the AP-HM register N° 2020-151 and 2020-152. 

## 3. Results

A total of 424 patients are described in the present report (as illustrated in Figure 1), and their epidemiological, clinical, and baseline ECG characteristics are presented in Table 3. All outpatients were patients with mild to moderate severity with a NEWS-2 score ≤ 5. The mean QTc duration was 396.8 ms with standard deviation of 29.4 ms. The results showed that 11 (2.6%) of the patients were not given HCQ-AZ treatment for cardiac reasons: 1 (0.2%) patient had a QTc of 480 ms and a T-wave pattern suggestive of long QT syndrome; 3 (0.7%) patients showed type I Brugada pattern; 5 (1.2%) patients had a known severe heart disease (2 with ischemic cardiomyopathies, 1 with hypertrophic cardiomyopathy, 1 with idiopathic cardiomyopathy, and 1 with valvular heart disease); 1 (0.2%) patient was suspected of having severe heart disease due to a severe non-specific intraventricular conduction delay; and 1 (0.2%) patient was receiving chronic treatment with amiodarone.

Within the total cohort (*n* = 424), 9 patients had an initial QTc over 460 ms (2.1%), and 3 of these patients were contraindicated due to an underlying cardiopathy (*n* = 2) or had repolarization pattern suggestive of long QT syndrome. Of the six patients who effectively received HCQ + AZM, four had a right bundle branch block. The last two patients presented neither a pattern of long QT syndrome nor an underlying cardiomyopathy or a family history of sudden cardiac death. Treatment was, therefore, approved in all six patients.

In consequence, 413 (97.4%) patients were prescribed HCQ-AZ for 5 days and HCQ for an additional 5 days. None of the patients reported palpitations suggestive of a malignant ventricular arrhythmia or syncope. Two patients died from non-sudden cardiac death at hospital (one due to Takotsubo syndrome and one due to acute respiratory distress). In both cases, their heart rhythm had been closely monitored by cardiac telemetry and did not show any signs of arrhythmia. None of the 6 patients with a basal QTc over 460 ms had either QTc prolongation or symptoms suggestive of ventricular arrhythmia under HCQ-AZ.

As shown in Table 4, in the treated patients, QTc was slightly but statistically significantly prolonged from baseline to 48 h, mainly driven by a QTc prolongation in women. Figure 2 shows the distribution of the QTc changes at Day 2 when compared with baseline. QTc prolongation was particularly observed in female outpatients <65 years without a cardiovascular disease (Table 4). A total of 53 patients (36 women) had a QTc prolongation of ≥30 ms. Gender was a risk factor as women more frequently had a QTc prolongation of ≥30 ms (OR, 2.17; 95% CI, 1.17 to 4.00; *p* = 0.01). Ten patients (six women) had a QTc prolongation of ≥60 ms, and none had a QTc > 500 ms. No clinical or electrocardiographic variables were statistically associated with QTc prolongation > 60 ms. At Day 2, heart rate was lower than at baseline (71.8 ± 12.5 beats/min vs. 74.6 ± 13.4 beats/min, respectively; *p* < 0.0001), and the QRS duration was significantly prolonged (87.7 ± 13.3 ms vs. 81.5 ± 15.3 ms, respectively; *p* < 0.0001). 

The randomly selected subsample of ECGs used for the comparison of manual QTc (cardiologist) and automatic QTc (recorder; Bazett formula) consisted of 200 ECGs from 109 women and 91 men. The automatic QTc was longer than the measured QTc (417.97 ± 24.83 ms for automatic vs. 391.67 ± 24.69 ms for manual; *p* < 0.0001). The correlation between the two measures was strong (correlation coefficient, 0.87; 95% CI, 0.83 to 0.90), but their agreement was moderate (intraclass correlation coefficient, 0.56; 95% CI, −0.08 to 0.84). The mean bias was 26.30 (95% CI, 24.54 to 28.06). Only one patient had an automatic QTc measurement that was shorter than the manual QTc measurement, with a difference of 1 ms.

## 4. Discussion

Our results were obtained from 424 SARS-CoV-2-infected patients with mild-to-moderate symptoms, who were candidates for a treatment combining HCQ-AZ for 5 days followed by HCQ alone for an additional 5 days, during the first weeks of COVID-19 patient care in our center. These results show that a pragmatic safety strategy can be implemented in an emergency setting to ensure that this treatment has an acceptable cardiac rhythm safety. Based on the first medical assessment, including a 12-lead ECG, 2.6% of the patients did not receive the treatment for cardiac reasons. Among the patients who were eligible, treatment with HCQ-AZ did not lead to QTc prolongation necessitating the interruption of treatment, and no sudden cardiac deaths were observed. Moreover, the automatic determination of QTc by the ECG recorder appeared to be consistently longer than that measured by a cardiologist, suggesting that it may be safe to use this value. Initial cardiac safety publications evaluating HCQ treatment in hospitalized patients with COVID-19, with a mean age over 60, a longer basal QTc interval (respectively, 396 ± 28.7 vs. 435 ± 24 ms, *p* < 0.001), and including severe and critically ill patients with COVID-19 have shown significant QTc lengthening in some patients [13,14], possibly leading to severe ventricular arrythmia. As shown in our cohort, a simple clinical evaluation with kalemia and the use of the first ECG enabled us to initiate the treatment with acceptable safety in terms of potential arrhythmias, as confirmed by the second ECG and the clinical outcomes of our patients. The significant, but modest, QTc lengthening observed here is within the range of what has been reported when HCQ is combined with AZ [15]. A slight but significant prolongation of QRS duration was also observed and is consistent with the known effects of chloroquine on electrocardiogram [21]. In any case, our results in a higher-risk population (i.e., initial QTc over 460 ms, severe and critically ill patients) have yet to be proven in a larger study.

HCQ is a derivative of chloroquine, which has similarities to quinine, and may, therefore, prolong QT interval, although the effect is expected to be modest [22]. AZ has a low affinity for hERG channel [16], and no proarrhythmia potential of the two drugs when used in combination has been shown at therapeutic doses [17]. 

We acknowledge that our monitoring strategy could have been stricter, but it should be borne in mind that these measures were implemented in the context of an ongoing and rapidly increasing pandemic, taking into account the balance between benefits and risks for patients. The strategy was consistent with safety guidelines issued by the American College of Cardiology [23], which recommended that intensity of QT and arrhythmia monitoring should be considered in the context of risk level, resource availability, and quarantine considerations. Our experience is in accordance with their proposal, which should reassure clinicians when using these drugs. Even in this context, relatively simple measures can be implemented, such as the ones we used, including warning colleagues about the added risk of combining medications that could lengthen QT interval or decrease kalemia; reminding colleagues on how to recognize, based on patients’ history and medications, whether there is a risk of channelopathy or severe heart disease; being especially cautious with the elderly, especially women who, as we confirm here, may be most at risk of significant prolongation of QTc interval. In addition, in this emergency situation, we feel that it is extremely important to establish a range of measures to facilitate “on-line” communication between the teams prescribing anti-infectious drugs and the cardiologists. Obviously, our results should not be extrapolated to situations in which these measures cannot be implemented. 

In a smaller cohort [24], we previously published daily monitoring of QTc under HCQ-AZ, using a smartwatch, in cases of early stage COVID-19 infections with mild to moderate symptoms. Although we did not find a high risk of QTc prolongation, we do not recommend using our strategy when more precise monitoring of QT interval may be required. In such cases, monitoring must be implemented as recommended [25] to allow patients to benefit from the treatment under the strictest safety conditions. This is particularly true in SARS-CoV-2-infected patients in intensive care, taking into account their increased risk of electrolyte disorders [26]. 

Most of the patients (78.5%) in our cohort were outpatients, and only 11.1% were aged over 65, with mild to moderate symptoms. Given that treatment is not toxic and has a simple management protocol, that the at-risk population could not be perfectly identified at baseline, and that early treatment is critical for efficacy, we have discussed elsewhere how this treatment should be initiated at the early stage of the disease [6]. 

Our results show that automatic QTc measurement by an ECG recorder leads to a mean systematic error that overestimates the QTc compared with manual assessment. It could, therefore, be acceptable, given the pandemic circumstances, to generalize automatic assessment of QTc at the initiation of HCQ-AZ treatment. This strategy may reduce the number of patients in whom a second opinion from a cardiologist is needed, e.g., when the automatic QTc exceeds 460 ms. 

Additionally, it should be reminded that the Bazett formula was used to estimate QTc, as this is the most common approach. This formula is suboptimal [27] and overestimates QTc, particularly when heart rate is high, which might often be the case in febrile patients. Obviously, this will increase the safety process. These points are important when QTc monitoring may be performed using surrogates for conventional ECG recorders [25]. 

Once again, no torsade de pointe was identified in the 424 included patients. Furthermore, in order to identify the COVID patients most at risk based on the present study, we found that HCQ significantly prolongs QTc in women who are less than 65 years of age, managed on an outpatient basis, and do not have a cardiovascular disease (Table 4). This is a population with an extremely low risk of torsade de pointes. In contrast, we did not find any significant QTc prolongation in male patients, those older than 65 years, and patients with a cardiovascular disease or were hospitalized (Table 4). Hence, in the population with more severe cardiac vulnerability, we found that the risk of torsade de pointe, based on QTc prolongation, was lower. Based on cardiologic expertise, selected patients should benefit from smartwatch electrocardiogram and artificial intelligence, demonstrated by our team as a relevant and modern approach [24]. Finally, in the literature, studies with careful and expert cardiac monitoring did not find sudden death and cardiac mortality related to HCQ-AZ treatment [28]. 

This study, which was performed during the first months of the pandemic but published almost three years later, does not aim to modify the management strategy of COVID-19. On the contrary, while the situation of COVID-19 has completely changed as well as its management, this study can clarify the toxicity that was initially alleged but was not observed in our center. This is important for future generations. Indeed, this report provide insights for clinicians and researchers that the toxicity of a repurposed treatment used in a new indication depends, to a large extent, on the use of a pragmatic, simple care protocol involving close collaboration between infectious disease specialists and relevant specialists (here, cardiologists). In this case, the availability of several latest-generation ECG machines with automatic interpretation (kindly provided by the cardiac rhythmology unit) and the setting up of a telephone hotline for urgent cardiological advice in case of doubt also contributed to the successful management and safe prescription of dual therapy with HCQ-AZ.

This study highlights an important point. In practice, patient interview, co-medications, ECG, and biochemistry are routine tests in most hospitals nowadays. A more precise and specific approach is, therefore, not necessary in practice. However, this suggests that basic medicine is critical during pandemics. A pragmatic approach is to identify patients who are at risk of dying from the disease and will benefit most from treatment. Among these patients, the next step is to identify those at risk for treatment side effects. In patients at risk for disease complications and treatment side effects, a complete pre-therapy workup is necessary. The practice of ECG is not always easy to be performed routinely by a general practitioner. During the COVID-19 pandemic, the performance of ECGs in a city for patients at risk but without meeting the severity criteria was often complicated. The results of the present study are in favor of making ECGs available to general practitioners, who are capable of interpreting normal ECGs (notably with the help of the machine). In case of detection of abnormality, a cardiological expert’s opinion would be desirable. This study is, therefore, an argument in favor of making ECG available to any general practitioner.

## 5. Conclusions

Starting in April 2020, the indications for Day 2 control ECG were restricted after an initial workup showing that all contraindicative repolarization abnormalities that had been detected on the first ECG had been resolved, and that HCQ-AZ could be used safely. This ECG monitoring was conducted in an emergency situation as an early response to the ongoing COVID-19 pandemic. This report does not aim to contribute to knowledge of the efficacy of treating COVID-19 with HCQ-AZ. Indeed, we have reported our 2020 data elsewhere [5,6]. More results will soon be discussed in the SARS-CoV-2 variant vaccination and post-vaccination era. Here, we wanted to focus on the so-called cardiac toxicity of HCQ-AZ combination. Our results indicate that the risks of severe arrhythmia induced by combined HCQ-AZ therapy for COVID-19, if any, can be minimized by a simple clinical management, including careful assessment of contraindications (mainly cardiological history, co-medications, kalemia, and initial ECG), interruption of other drugs prolonging QTc if possible, and correction of hypo- or hyperkalemia. This has been confirmed by a previous study presenting data on the treatment of more than 30,000 patients in our center in 2020 and 2021, including 4000 hospitalized patients with moderate to severe symptoms [29]. If this simple and systematic cardiac rhythm safety evaluation is performed, and contraindications are monitored, treatment with HCQ-AZ in early stage COVID-19 is safe and is not associated with clinically relevant cardiac rhythm side effects. In patients taking only one QT-prolonging drug (such as AZ alone) and in the absence of other risk factors for torsade de pointes, routine assessment of QT interval does not appear to be recommended. This work provides evidence that QT-prolonging anti-infective drugs (Table 1) can be used safely in acute life-threatening infections, provided that a strict protocol and close collaboration between infectious disease specialists and rhythmologists are followed.

The present work provides real-world data on the safety of prescribing drugs that have been known for several decades. The evaluation of the tolerance of repurposed drugs in new indications is part of the advancement of knowledge. This is essential because these drugs will be the first available for new life-threatening diseases in the future. It is critical to ensure that physicians do not leave unprepared when managing future patients in the next deadly pandemic. The main message of the present article is that the safety of a QT-prolonging therapy can be ensured in a simple way based on a few pragmatic measures and the performance of an ECG. As many anti-infective agents are QT-prolonging drugs (Table 1), the present study is useful for the development of strategies for future infectious diseases and pandemics in the new millennium.

## Figures and Tables

**Figure 1 medicina-59-00863-f001:**
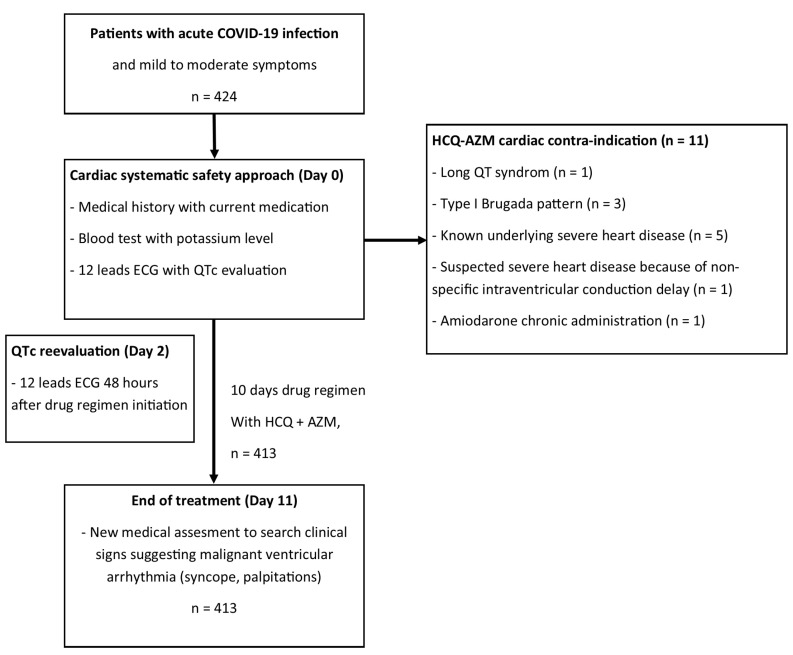
Flowchart.

**Figure 2 medicina-59-00863-f002:**
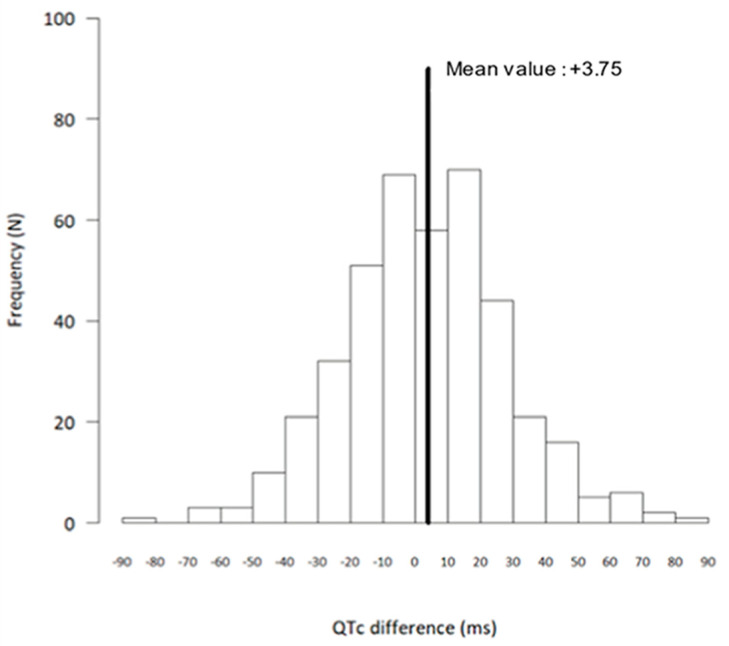
Distribution of differences in corrected QT interval (QTc): Day 2 versus baseline, showing absolute differences between Day 2 and baseline QTc (*n* = 413 patients).

**Table 1 medicina-59-00863-t001:** QT-prolonging anti-infective drugs ^a^.

Molecules	Indications in Infectious Diseases
**Polyenes** Amphotericin B	Fungal infections
**Antimalarial combinations** Artemether/lumefantrine Artenimol/piperaquine	Malaria
**HIV Protease inhibitors** Atazanavir Lopinavir/ritonavir Nelfinavir Saquinavir	HIV
**Macrolides** Azithromycin ^b^ Clarithromycin Erythromycin Roxithromycin Telithromycin	Bacterial infections
**Diarylquinolines** Bedaquiline	Tuberculosis
**Antimalarial quinolines** Chloroquine Hydroxychloroquine ^b^ Primaquine Quinine	Malaria, Q fever
**Quinolones** Ciprofloxacin Levofloxacin Moxifloxacin Norfloxacin Ofloxacin	Bacterial infections
**Leprostatics** Clofazimine	Mycobacterial infection
**NNRTIs** Efavirenz	HIV
**Azole antifungals** Fluconazole Posaconazole Voriconazole	Fungal infections
Meglumine antimoniate	Leishmaniosis
**Amebicides and miscellaneous antibiotics** Metronidazole	Amoebiasis and anaerobic bacterial infections
**Antipseudomonal penicillins** Piperacillin/tazobactam	Bacterial infections
**Purine nucleosides** Remdesivir	COVID-19
**Sulfonamides** Sulfamethoxazole and trimethoprime	Bacterial and parasitic infections

NNRTIs: non-nucleoside reverse transcriptase inhibitors; HIV: human immunodeficiency virus. ^a^ This list was extracted from https://crediblemeds.org/, accessed on 6 April 2023. ^b^ Hydroxychloroquine and azithromycin are the two molecules included in our COVID-19 standard treatment protocol. **Bold**: pharmacological class. This list is not exhaustive.

**Table 2 medicina-59-00863-t002:** Simple and systematic cardiac rhythm safety evaluation proposed before the initiation of HCQ-AZ in 2020 during the first month of COVID-19 patient care at the IHU Méditerranée Infection.

Item	Contraindication ^a^	Safe Prescription of HCQ-AZM
Patient interview: cardiac history	Severe structural or electrical heart disease ^a^	- Absence of history of severe cardiopathy associated with increased risk of torsade de pointe
Patient interview: co-medications	Co-medication of HCQ-AZ with QTc-prolonging drugs ^b^	- Patients without QTc-prolonging drugs- Patients for whom QT-prolonging therapy could be discontinued for 10 days
Initial 12-lead ECG	Baseline corrected QT interval > 500 ms, channelopathy, Brugada syndrome, pathological Q waves, left ventricular hypertrophy, and left bundle branch block.Any abnormal ECG after cardiological advice ^c^	- Patients with normal ECG- Patients with abnormal ECG but no contraindication to HCQ-AZ after cardiological advice ^c^
Kalemia	Dyskalemia (K+ <3.6 mmol/L or K+ >5 mmol/L) ^d^	- Patients without dyskalemia- Patients with corrected dyskalemia

^a^ Outside QT-prolonging drugs and predefined ECG abnormalities (baseline corrected QT interval > 500 ms, channelopathy, Brugada syndrome, pathological Q waves, left ventricular hypertrophy, and left bundle branch block); cardiological contraindications should be systematically confirmed by a cardiologist’s advice. ^b^ List of QT-prolonging drugs can be easily assessed using online databases, such as https://www.crediblemeds.org (accessed on 31 March 2023). ^c^ Photography of abnormal ECG should be systematically sent to a cardiologist. ^d^ In these patients, treatment could be initiated as soon as kalemia is corrected.

**Table 3 medicina-59-00863-t003:** Patient characteristics.

Characteristic	Value (*n* = 424)
Male sex—no. (%)	208 (49.5)
Mean age ± SD—year	46.3 ± 16.1
≥65 years—no. (%)	47 (11.1)
Clinical setting—no. (%)	
Day-care patients	333 (78.5)
Inpatients	91 (21.5)
Cardiovascular treatment—no. (%)	
ACE inhibitors/ARBs	34 (8.0)
Beta-blockers	15 (3.5)
Diuretics	17 (4.0)
Calcium channel blockers	1 (0.2)
Digoxin	1 (0.2)
Flecainide	4 (0,9)
Amiodarone	1(0.2)
Baseline ECG	
Mean heart rate ± SD—beats/min	74.6 ± 13.6
Mean QRS duration ± SD—ms	82.3 ± 1646
Mean QTc duration ± SD—ms	396.8 ± 29.4
Initial ECG patterns suggesting:	
Long QT interval—no. (%)	1 (0.2)
Type I Brugada syndrome—no. (%)	3 (0.7)
Bundle branch block—no. (%)	40 (9.4)
Left ventricular hypertrophy—no. (%)	4 (0.9)
Pathological Q waves—no. (%)	9 (2.1)
Early repolarization pattern—no. (%)	37 (8.7)
QTc risk score Tisdale score (points), median (IQR)	7 (6–7)

ACE: angiotensin-converting enzyme; ARB: angiotensin II receptor blocker: ECG: electrocardiogram; and SD: standard deviation.

**Table 4 medicina-59-00863-t004:** Corrected QT interval (QTc) values obtained at baseline and at Day 2, and their comparison.

Variable	Mean Baseline QTc ± SD—ms	Mean Day 2 QTc ± SD—ms	Mean Absolute Difference in QTc (Day 2 vs. Baseline) ± SD—ms	*p*-Value for Comparison of QTc between Baseline and Day 2
General population (*n* = 413)	396.0 ± 28.7	399.7 ± 28.7	+3.75 ± 25.4	0.003
**Sex**				
Female (*n* = 214)	401.1 ± 27.4	407.0 ± 25.6	+5.61 ± 25.3	0.001
Male (*n* = 199)	390.2 ± 29	392.0 ± 29.8	+1.73 ± 25.5	0.31
**Age**				
<65 years (*n* = 366)	393.7 ± 27.4	397.2 ± 27.8	+3.56 ± 25.3	0.007
≥65 years (*n* = 47)	413.9 ± 32.2	419.1 ± 27.8	+4.62 ± 26.6	0.19
**Cardiovascular disease**				
Absent (*n* = 350)	392.2 ± 27.2	396.3 ± 28.0	+4.04 ± 25.9	0.004
Present (*n* = 63)	416.8 ± 27.9	418.9 ± 24.5	+2.11 ± 22.7	0.47
**Patient setting**				
Day-care (*n* = 328)	391.8 ± 27.8	395.9 ± 27.8	+4.11 ± 26.2	0.005
Inpatient (*n* = 85)	412.3 ± 26.3	414.6 ± 27.4	+2.33 ± 22.1	0.33

SD denotes standard deviation. **Bold**: Classes of variables.

## Data Availability

Data are available upon reasonable request.

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
