# Peer review of "Cardiovascular Safety of Hydroxychloroquine–Azithromycin in 424 COVID-19 Patients"

_medicina, 2023, doi:10.3390/medicina59050863_

Round 1

Reviewer 1 Report

Simple, observational, methodologically sound article.

- Would you routinely recommend QT assessment to all patients who are to take antibiotic therapy with azithromycin independently of other medications?

- do you have the standard deviation data of the QTc measurement?

- Can you report qtc according to Bazzett and Fridericia correction?

- Do you have any data regarding the intake of azithromycin with respect to the meal?

Author Response

Reviewer 1 Comments and Suggestions for Authors

Simple, observational, methodologically sound article.

Question 1. Would you routinely recommend QT assessment to all patients who are to take antibiotic therapy with azithromycin independently of other medications?

Answer: We thank the reviewer for this very pertinent comment. We add the following sentence line 338:

“In patients taking only one QT-prolonging drug (such as AZ alone) and in the absence of other risk factors for torsades de pointes, routine assessment of the QT interval does not appear to be recommended.”

Question 2. do you have the standard deviation data of the QTc measurement?

Answer: We thank the reviewer for this helpful comment. We add the following sentence line 154:

“The mean QTc duration was 396.8 ms with standard deviation of 29.4 ms”

Question 3. Can you report qtc according to Bazzett and Fridericia correction?

Answer: The Bazzett correction was used. While the Fridericia is also useful, it was not assessed in the present study. The Bazzett correction is recognized as a valid method for QT correction. We mention in the discussion line 274:

“Also, it should be reminded that the Bazett formula was used to estimate the QTc, as this is the most common approach. This formula is suboptimal [27] and overesti-mates the QTc, particularly when the heart rate is high, which might often be the case in febrile patients. Obviously, this will increase the safety process. These points are important when QTc monitoring may be performed using surrogates for conventional ECG recorders [25].”

Pertinent references

  1. Roden Dan M., Harrington Robert A., Poppas Athena, Russo Andrea M. Considerations for drug interactions on QTc in ex-ploratory COVID-19 treatment. Circulation 2020, 141, e906‑7.
  2. Vandenberk B, Vandael E, Robyns T, Vandenberghe J, Garweg C, Foulon V, et al. Which QT correction formulae to use for QT monitoring? J Am Heart Assoc 2016, 5, e003264.

Question 4. Do you have any data regarding the intake of azithromycin with respect to the meal?

Answer: We thank the reviewer for this helpful comment. While we do not have this information, prescription of azithromycin was 250mg 2 pills the first day and 250mg 1 pill during meal. We add this information in the manuscript line 118.

Reviewer 2 Report

Thank you for submitting your manuscript titled "Cardiovascular Safety of Hydroxychloroquine-Azithromycin in 424 COVID-19 patients." I offer the following comments:

Firstly, I would like to address the relevance of the manuscript. Given the current COVID-19 scenario and the results from randomized controlled trials proving the ineffectiveness of hydroxychloroquine / azithromycin for COVID-19, it is important to consider the current relevance of the manuscript.

Had the manuscript been written two years ago, it may have been more relevant at that time. With the rapid changing landscape of the COVID-19, current relevance of the manuscript is doubtful

While the manuscript presents a simple safety approach before initiating HCQ/AZ for patients with COVID-19, it does not contribute anything significant to the existing knowledge.

Moreover, the safety evaluation performed in the study, - Patient interview: cardiac history, Patient interview: co-medications, ECG, Biochemistry, is a routine screen in most hospitals nowadays. A more precise and specific monitoring approach could have been better and more rational.

It would be helpful if the authors add the sampling method used for the study and clarifiy the details of statistical analysis carried out in the manuscript.

Additionally, the manuscript need  an in-depth discussion of the limitations of the study, given the current changed landscape of COVID-19 situation and its management.

Regarding the results, the authors have mentioned

“The predictive effect of various characteristics (age, heart rate, sex, and cardiopathy) on QTc prolongation ≥ 30 ms and ≥ 60 ms was assessed by estimating odds ratios (ORs) with 95% confidence intervals (CIs).”

“Gender 177 was a risk factor as women had more frequently a QTc prolongation of ≥ 30 ms (OR, 2.17; 178 95% CI, 1.17 to 4.00; P=0.01).”

The manuscript would benefit from the addition of regression model results related to odds ratios in a tabular form for clarity and interpretation.

Author Response

Reviewer 2.

Thank you for submitting your manuscript titled "Cardiovascular Safety of Hydroxychloroquine-Azithromycin in 424 COVID-19 patients." I offer the following comments:

Question 5. Firstly, I would like to address the relevance of the manuscript. Given the current COVID-19 scenario and the results from randomized controlled trials proving the ineffectiveness of hydroxychloroquine / azithromycin for COVID-19, it is important to consider the current relevance of the manuscript.

Answer: The reviewer is absolutely relevant. Thanks to his provocative speech, he allows us to highlight even better the approach developed in our center. We believe that the situation of the COVID pandemic is a very good case study for future pandemics. The health crisis has been accompanied by a questioning of the place of gold standard for randomized trials. While randomized trials are considered the gold standard for scientific evidence in medicine, the question is: are there situations where randomized trials can be flawed? Are they suitable for evaluating complex management including recycled drugs as early as possible? This is a real relevant question that will certainly be the subject of intense research in the years to come, especially with the advent of precision medicine, real world data, big data and artificial intelligence. Thanks to the reviewer, we will be researching this topic but it is not the focus of this article. So we added the following paragraph at the end of the conclusion and MS line 344:

" The present work provides real-world data on the safety of prescribing drugs that have been known for several decades. The evaluation of the tolerance of repurposed drugs in new indications is part of the advancement of knowledge. This is essential because these drugs will be the first available for new life-threatening diseases in the future. It is critical to ensure that the physicians of tomorrow do not leave unprepared to manage future patients in the next deadly pandemic. The main message of the pre-sented article is that the safety of a QT-prolonging therapy can be ensured in a simple way by a few pragmatic measures and the performance of an ECG. As many QT-prolonging drugs are used in infectious diseases (Table 1), the present study is useful for the development of strategies for future infectious diseases and pandemics for the new millenium.”

Also, to better emphasize the generalizability of this safety approach to future situations, the Table mentioning important antimicrobial drugs associated with prolonged QTc have been moved and is now cited as soon as the end of the introduction. We added the following sentence line 69:

“However, specific cardiovascular safety was not reported in our center. This is im-portant to decipher if a strict standardized protocol for an association of QT prolonging drugs allow safe prescription for acute infectious disease. Indeed, many important an-timicrobial drugs are associated with prolonged QTc (Table 1).”

Question 6. Had the manuscript been written two years ago, it may have been more relevant at that time. With the rapid changing landscape of the COVID-19, current relevance of the manuscript is doubtful

Answer: We understand the questioning of the reviewer. However, the relevance of the present manuscript lies in showing for the record that the management of a novel fatal disease must be based on medical knowledge. As published, simple early management with home screening for happy hypoxia divided the mortality rate by 4 in a German study unrelated to our center (Lim, 2022). Basic medical knowledge should not be neglected in the future in front of unknown diseases. Accordingly, we added line 344:

“The present work provides real-world data on the safety of prescribing drugs that have been known for several decades. The evaluation of the tolerance of repurposed drugs in new indications is part of the advancement of knowledge. This is essential because these drugs will be the first available for new life-threatening diseases in the future. It is critical to ensure that the physicians of tomorrow do not leave unprepared to manage future patients in the next deadly pandemic. The main message of the pre-sent article is that the safety of a QT-prolonging therapy can be ensured in a simple way by a few pragmatic measures and the performance of an ECG. As many an-ti-infective agents are QT-prolonging drugs (Table 1), the present study is useful for the development of strategies for future infectious diseases and pandemics for the new millenium.”

References : Lim A, Hippchen T, Unger I, et al. An Outpatient Management Strategy Using a Coronataxi Digital Early Warning System Reduces Coronavirus Disease 2019 Mortality. Open Forum Infect Dis. 2022;9(4):ofac063. Published 2022 Feb 8. doi:10.1093/ofid/ofac063

Question 7. While the manuscript presents a simple safety approach before initiating HCQ/AZ for patients with COVID-19, it does not contribute anything significant to the existing knowledge.

            Answer: This has been addressed in Question 6.

Question 8. Moreover, the safety evaluation performed in the study, - Patient interview: cardiac history, Patient interview: co-medications, ECG, Biochemistry, is a routine screen in most hospitals nowadays. A more precise and specific monitoring approach could have been better and more rational.

Answer: The reviewer’s comment is highly relevant. Accordingly, we added line 305:

“The study highlights an important point. In practice, patient interview, co-medications, ECG, biochemistry are routine tests in most hospitals nowadays. A more precise and specific approach is therefore not necessary in practice. However, this suggests that basic medicine is critical during pandemics. A pragmatic approach: identify patients at risk of dying from the disease who will benefit most from treat-ment. Among these patients, identify those at risk for treatment side effects. In patients at risk for disease complications and treatment side effects, a complete pre-therapy workup is necessary. The practice of ECG is not always easy to be performed routinely today by the general practitioner. During the pandemic, the performance of ECGs in the city for patients at risk but without severity criteria was often complicated. The results of the present study are in favour of making ECGs available to general practi-tioners in the city, who are capable of interpreting a normal ECG (notably with the help of the machine). In case of detection of abnormality, a cardiological expert opinion in town would be desirable. This study is therefore an argument in favor of making the ECG available to any general practitioner.”

Question 9. It would be helpful if the authors add the sampling method used for the study and clarifiy the details of statistical analysis carried out in the manuscript.

Answer: The sampling method used for the study has been detailed. Subjects included were the first patients treated in our center. This has been added line 83:

Patients were the first adults seen for SARS-CoV-2 infection at the Institut Hos-pitalo-Universitaire (IHU) Méditerranée Infection, France, between March 3, 2020, and April 5, 2020.”

And statistical analyses have been detailed line 132:

“Quantitative variables are presented as means ± standard deviations and categorical variables as numbers (percentages). An analysis was conducted in patients for whom both the baseline ECG and the Day 2 ECG were available. Initial QTc and Day 2 QTc were compared by means of a paired T-test, in the overall population and ac-cording to subgroups (age, sex, cardiopathy, and hospitalisation). The predictive effect of various characteristics (age, heart rate, sex, and cardiopathy) on QTc prolongation ≥ 30 ms and ≥ 60 ms was assessed by estimating odds ratios (ORs) with 95% confidence intervals (CIs).

An analysis was conducted in a random subsample of the overall population, to compare the cardiologist’s interpretation and the automatic interpretation of the QTc. Correlation and agreement between measures were assessed by estimating the corre-lation and the intraclass correlation coefficient (ICC), respectively, with 95% CI. All analyses were performed using R software, version 3.6.3. All tests were two-sided, and P values < 0.05 were considered to be statistically significant.”

Question 10. Additionally, the manuscript need  an in-depth discussion of the limitations of the study, given the current changed landscape of COVID-19 situation and its management.

Answer: We agree with the reviewer’s comment. In consequence, we added line 305:

“This study, performed during the first months of the pandemic, but published almost 3 years later, does not aim to modify the management strategy of COVID-19. On the contrary, while the situation of COVID-19 has completely changed as well as its management, this study deserves to be published to clarify the toxicity initially alleged but finally not observed in our center. This is important for future generations. Indeed, this report will allow them to understand that the toxicity of a repurposed treatment used in a new indication depends to a large extent on the respect of a pragmatic, simple care protocol involving close collaboration between infectious diseases specialists and specialists in possible and/or supposed toxicity (here, cardiologists). In this case, the availability of several latest generation ECG machines with automatic interpretation (kindly provided by the cardiac rhythmology unit) and the setting up of a telephone hotline for urgent cardiological advice in case of doubt also contributed to the successful management and safe prescription of dual therapy with HCQ-AZ.”

Question 11. Regarding the results, the authors have mentioned “The predictive effect of various characteristics (age, heart rate, sex, and cardiopathy) on QTc prolongation ≥ 30 ms and ≥ 60 ms was assessed by estimating odds ratios (ORs) with 95% confidence intervals (CIs).”

“Gender was a risk factor as women had more frequently a QTc prolongation of ≥ 30 ms (OR, 2.17; 178 95% CI, 1.17 to 4.00; P=0.01).”

The manuscript would benefit from the addition of regression model results related to odds ratios in a tabular form for clarity and interpretation

Answer: We thank the reviewer for his helpful contribution. This was not a multi-variable model. The analysis of different co-variables associated with QTc prolongation was detailed in Table 3. We added accordingly the following sentence in the abstract line 25:

“QTc prolongation was particularly observed in female outpatients < 65 years without cardiovascular disease.”

And in the results line 186:

“QTc prolongation was particularly observed in female outpatients < 65 years without cardiovascular disease (Table 3).”
